# Neonatal Vitamin D and Associations with Longitudinal Changes of Eczema up to 25 Years of Age

**DOI:** 10.3390/nu16091303

**Published:** 2024-04-26

**Authors:** Rong Zeng, Caroline J. Lodge, Jennifer J. Koplin, Diego J. Lopez, Bircan Erbas, Michael J. Abramson, Darryl Eyles, Anne-Louise Ponsonby, Matthias Wjst, Katrina Allen, Shyamali C. Dharmage, Adrian J. Lowe

**Affiliations:** 1Allergy and Lung Health Unit, Melbourne School of Population and Global Health, University of Melbourne, Melbourne, VIC 3052, Australia; rzzeng@student.unimelb.edu.au (R.Z.); clodge@unimelb.edu.au (C.J.L.); diego.lopezperalta@unimelb.edu.au (D.J.L.); lowea@unimelb.edu.au (A.J.L.); 2Murdoch Children’s Research Institute, Melbourne, VIC 3052, Australia; annelouise.ponsonby@florey.edu.au (A.-L.P.); katie.allen@mcri.edu.au (K.A.); 3Centre for Food and Allergy Research, Murdoch Children’s Research Institute, Melbourne, VIC 3052, Australia; 4Child Health Research Centre, University of Queensland, Brisbane, QLD 4072, Australia; j.koplin@uq.edu.au; 5School of Psychology and Public Health, La Trobe University, Melbourne, VIC 3086, Australia; b.erbas@latrobe.edu.au; 6Violet Vines Marshman Centre for Rural Health Research, La Trobe University, Bendigo, VIC 3550, Australia; 7School of Public Health and Preventive Medicine, Monash University, Melbourne, VIC 3004, Australia; michael.abramson@monash.edu; 8Queensland Brain Institute, University of Queensland, Brisbane, QLD 4072, Australia; d.eyles@uq.edu.au; 9Queensland Centre for Mental Health Research, University of Queensland, Brisbane, QLD 4076, Australia; 10Florey Institute of Neuroscience and Mental Health, University of Melbourne, Melbourne, VIC 3010, Australia; 11Institut für Medizinische Informatik, Statistik und Epidemiologie, Technische Universität München, 80333 München, Germany; wjst@tum.de

**Keywords:** 25-hydroxyvitamin D, dried whole blood spots, sensitization, eczema phenotype, risk factors

## Abstract

Background: Early-life vitamin D is a potentially modifiable risk factor for the development of eczema, but there is a lack of data on longitudinal associations. Method: We measured 25(OH)D3 levels from neonatal dried blood spots in 223 high-allergy-risk children. Latent class analysis was used to define longitudinal eczema phenotype up to 25 years (4 subclasses). Skin prick tests (SPTs) to 6 allergens and eczema outcomes at 6 time points were used to define eczema/sensitization phenotypes. Associations between 25(OH)D3 and prevalent eczema and eczema phenotypes were assessed using logistic regression models. Results: Median 25(OH)D3 level was 32.5 nmol/L (P25-P75 = 23.1 nmol/L). Each 10 nmol/L increase in neonatal 25(OH)D3 was associated with a 26% reduced odds of early-onset persistent eczema (adjusted multinomial odds ratio (aMOR) = 0.74, 95% CI = 0.56–0.98) and 30% increased odds of early-onset-resolving eczema (aMOR = 1.30, 95% CI = 1.05–1.62) when compared to minimal/no eczema up to 12 years. Similar associations were seen for eczema phenotype up to 25 years. We did not see any strong evidence for the association between neonatal 25(OH)D3 and prevalent eczema or eczema/sensitization phenotype. Conclusions: Higher neonatal 25(OH)D3 levels, a reflection of maternal vitamin D levels in pregnancy, may reduce the risk of early-onset persistent eczema.

## 1. Introduction

Eczema (atopic dermatitis) is a highly prevalent itchy inflammatory skin condition [1]. While some studies have suggested that skin barrier dysfunction and immunologic disturbance (IgE-mediated sensitization caused by T_h_2 immune response) may lead to eczema [2,3], the underlying mechanisms and risk factors are not completely known. Vitamin D in early life seems to be an important factor in the developmental pathway of eczema. Both vitamin D deficiency and eczema are more prevalent in areas further from the equator [4] that have lower levels of ultraviolet sunlight. Vitamin D receptors (VDRs) are found in multiple immune cell types, and the complex of vitamin D and VDR appears to have a role in regulating immune responses [5]. Vitamin D may maintain a T_h_1/T_h_2 balance and suppress T_h_17, therefore enhancing the expression of IL-10 and inhibiting the production of IgE, which ultimately inhibits inflammation in the skin and decreases sensitization [6,7]. Given that the majority of eczema appears initially in the first year of life [8,9], it is possible that vitamin D in very early life may play a critical role in immune maturation and risk of developing eczema and allergic diseases [10].

Our recent systematic review and meta-analysis of six cohort studies [11,12,13,14,15,16] found evidence that higher levels of cord blood vitamin D were associated with a decreased risk of prevalent eczema in early childhood [17]. We observed consistent evidence that for every 10 nmol/L increase in neonatal vitamin D, the risk of eczema was reduced by approximately 11%. Maternal vitamin D level during pregnancy is a key driver of neonatal vitamin D [18], and vitamin D supplementation during pregnancy increases the neonatal vitamin D level [19]. Despite this, clinical trials of maternal vitamin D supplementation during pregnancy have not shown consistent protective effects [20,21,22,23], possibly due to lack of power, and/or contamination of the control group. Current antenatal care guidance in Australia [24], America [25], and by WHO [26] do not recommend the routine use of vitamin D supplementation during pregnancy, due to limited evidence on the benefits or harms of this intervention, especially regarding allergic diseases in the offspring.

Recently, it has been suggested that there are different childhood eczema phenotypes, based on when symptoms occur or concomitant sensitization, which may have different underlying causes [27,28]. Yet there were no existing studies that have examined the associations between neonatal vitamin D and longitudinal eczema or eczema/sensitization phenotypes from infancy to young adulthood. Therefore, we aimed to investigate the association between neonatal serum 25(OH)D3 levels (the most stable circulating form of vitamin D [29]) and prevalent eczema at age 1, 2, 6, 12, 18, and 25, and eczema phenotypes (including longitudinal eczema patterns and eczema/sensitization subclasses) up to 25 years of age.

## 2. Methods

### 2.1. Study Population

The Melbourne Atopy Cohort Study (MACS) is a longitudinal birth cohort based in Melbourne (latitude 37.8° S), Australia. A total of 620 infants, who had at least one first-degree family member with a history of allergic diseases (self-reported asthma, eczema, allergic rhinitis, and/or severe food allergy), were recruited prenatally between 1990 and 1994 [30]. MACS started as a randomized controlled trial exploring the effect of three infant formulas at weaning on the incidence of allergic diseases and has continued as a birth cohort [30].

### 2.2. Data Collection

An allergy-trained nurse conducted a telephone survey with the parents of the MACS participants every 4 weeks until 64 weeks, then at 18 months, and yearly at ages 2–7 years [30]. The participants were invited to complete a survey on allergic disease symptoms and attend clinical examinations at ages 12, 18, and 25 years. Skin prick testing (SPT) was performed at 6 months and 1, 2, 12, 18, and 25 years.

### 2.3. Assessment of Neonatal 25(OH)D3

At the 18-year follow-up, participants were asked to consent to the MACS study team accessing their newborn Guthrie screening cards (dried blood spots, DBS), collected within 48–72 h after birth via heel prick [31], to allow measurement of neonatal vitamin D levels. Levels of 25-hydroxyvitamin D3 (25(OH)D3) were measured using liquid chromatography–tandem mass spectrometry (LC/MS) from established methods [32]. As 25(OH)D3 is completely excluded from erythrocytes, we corrected DBS 25(OH)D3 concentration to serum 25(OH)D3 levels using the following formula: Serum 25(OH)D3 nmol/L = DBS 25(OH)D3 nmol/L/[1 − hematocrit] [32].

### 2.4. Eczema Definitions

Prevalent eczema in the first 2 years of life was defined as a parental report of either a doctor’s diagnosis of eczema or any rash (excluding scalp or nappy rashes) treated with topical steroids [30]. Prevalent eczema was defined as a parental report (from age 2 to 12 years) or self-report (at ages 18 and 25 years) of one or more episodes of eczema, a history of eczema with a rash treated with topical steroids, or episodes of eczema, which required a visit to the doctor, in the past 12 months.

Longitudinal eczema phenotypes from birth to 12 years were previously defined using latent class analysis [28], a statistical modeling procedure to identify latent subpopulations within a sample based on observed variables [33]. Five eczema subclasses were identified: early-onset persistent eczema, early-onset-resolving eczema, mid-onset persistent eczema, mid-onset-resolving eczema, and minimal/no eczema. We used the same statistical method to extend these phenotypes to age 25 [28]. For the best-fitting model, we considered the following criteria: The model that had the lowest Bayesian information criterion (BIC), the largest entropy value (0.87), and the smallest class membership was greater than 10% (Appendix A). The procedure resulted in a four-class model: early-onset persistent eczema (9.9%), early-onset-resolving eczema (10.5%), mid-onset persistent eczema (15.4%), and minimal/no eczema (64.2%, Appendix A). 

Eczema/sensitization phenotypes were based on participants’ eczema and food/aero allergen sensitization for up to 25 years for each time point. The 6 allergens were cow’s milk, peanut, egg white, house dust mite, ryegrass, and cat dander (Bayer, Spokane, WA, USA) performed on all occasions. Sensitization was defined by a positive skin prick test (SPT), i.e., a wheal ≥2 mm in the first 2 years of life or ≥3 mm at 12, 18, and 25 years to at least one of the above 6 allergen extracts. Participants were allocated into four eczema/sensitization subclasses at each age when SPT was performed: atopic eczema (eczema and positive SPT), non-atopic eczema (eczema but negative SPT), asymptomatic-sensitized (having no eczema but with positive SPT), or asymptomatic (no eczema and negative SPT).

### 2.5. Statistical Analysis

Logistic regression models were fitted to explore the associations between neonatal 25(OH)D3 levels and eczema. Results were reported as odds ratios (ORs) per 10 nmol/L increase in 25(OH)D3 and their 95% confidence intervals (95% CIs). Generalized estimation equations (GEEs) were used to combine measures of the prevalent eczema outcome for an individual participant over time. Interactions with time were assessed, using likelihood ratio tests, to determine if the associations changed over time. If the *p* for this interaction was <0.1, a pooled result was not reported. Multinomial logistic regression models were used for longitudinal eczema and eczema/sensitization phenotypes. 

A directed acyclic graph (DAG) was used to guide the selection of confounders (Appendix A). Based on the DAG presented in Appendix A (Model 1), we developed models adjusting for dog ownership, maternal factors (eczema/food allergy/hay fever history, age at delivery, birthplace, smoking during pregnancy), the presence of older siblings, and parental education. We also conducted a sensitivity analysis, which additionally adjusted for the season of birth (Appendix A, Model 2). 

The potential effect modification by early-life factors was explored using likelihood ratios or Wald tests (for GEE models). Factors tested were dog ownership, maternal eczema history, maternal prenatal smoking, mother’s birthplace, and parental education. Strata-specific results were reported if the *p* values for the interactions tested were <0.1 and an association (*p* < 0.05) was observed in one or other of the corresponding strata. Non-linearity of associations was explored using fractional polynomials, but we did not find any evidence for nonlinearity. All analyses were performed by using Stata release 16.0 (StataCorp, College Station, TX, USA).

### 2.6. Ethics

The initial phase of the study was approved by the Human Research Ethics Committee of the Mercy Hospital for Women up to 12-year follow-up (HREC R07/20 and R88/06). The 18- and 25-year follow-ups were approved by the University of Melbourne and the Royal Children’s Hospital Ethics Committees (HREC 28035 and 35179 on 17 December 2015), and the use of Newborn Screening Cards was approved by the director of Victorian Clinical Genetics Services (VCGS). Written informed consent was obtained from parents at recruitment and participants (at 18 and 25 years), including for release of the Newborn Screening Cards. 

## 3. Results

### 3.1. Study Population

As previously described, the parents of the MACS cohort were predominantly Australian-born and had relatively high socio-economic status (SES) [30]. Of the 620 participants, 223 (36%) had newborn vitamin D levels available. Those with a neonatal vitamin D measure differed in several ways from those without a vitamin D measure, including mothers being slightly older when they delivered (31.8 ± 4.3 vs. 30.9 ± 4.4 years old), fathers with a university degree or higher (71.9% vs. 55.2%), and mothers having never smoked during pregnancy (81.2% vs. 71.4%). However, we did not see a substantial difference in eczema status at any of the six time points between those with and without vitamin D measures. The peak prevalence of current eczema was in infancy (31.2%) for those with vitamin D measures, and it declined to 15.9% at age 6 and increased to 26.0% at age 25. Other characteristics including sex, season of birth, and family history of atopy were similar between these groups (Appendix A). 

### 3.2. Neonatal Vitamin D Status

The median sera-corrected 25(OH)D3 level for the 223 dried blood spot samples was 32.5 nmol/L (P25 = 21.6 nmol/L, P75 = 44.7 nmol/L, mean = 35.9 ± 18.6 nmol/L, Figure 1). As expected, there was a strong association between vitamin D and the season of birth (*p* < 0.001, Appendix A), with children born in winter having lower 25(OH)D3 levels. 

### 3.3. Neonatal Vitamin D Levels and 12-Month Prevalent Eczema at Age 1, 6, 12, 18, or 25 Years

The sensitivity analysis (additionally adjusted for the season of birth) showed an association between higher neonatal vitamin D levels and reduced risk of eczema at 1–2 years of age (Table 1). We explored the potential effect modification by the season of birth to the associations between vitamin D and prevalent eczema but did not have the power. 

Of the factors explored, only maternal smoking during pregnancy modified the association between vitamin D and prevalent eczema (Appendix A). For participants whose mothers had a history of smoking, there was a reduced risk of eczema at age 2 with higher neonatal vitamin D levels (OR = 0.78, 95% CI = 0.63–0.96 per 10 nmol/L increase in 25(OH)D3), while this association was not observed in the participants with non-smoking mothers (*p*-value of interaction = 0.04). 

### 3.4. Neonatal Vitamin D and Longitudinal Eczema Phenotypes up to 12 and 25 Years 

Compared to the minimal/no eczema subclass, higher levels of neonatal vitamin D were associated with a reduced risk of early-onset persistent eczema (aMOR = 0.74, 95% CI = 0.56–0.98) but higher odds of early-onset-resolving eczema (aMOR = 1.30, 95% CI = 1.05–1.62, Table 2) up to age 12 years. Sensitivity analysis, with additional adjustment for season of birth, did not materially change the results (Table 2). Similar trends of associations were seen for the eczema phenotypes from birth to 25 years of age (Table 3). 

Mother’s birthplace modified the association between vitamin D and eczema phenotypes up to 12 and 25 years (Appendix A). For children with mothers born in countries other than Australia and New Zealand (23 born in Europe/America, 3 born in Asia, and 2 in other countries), higher neonatal vitamin D levels were associated with decreased risk of early-onset persistent eczema up to 12 years (aMOR = 0.27, 95% CI = 0.11–0.68, Appendix A) compared to minimal/no eczema subclass. However, we did not observe an association for children whose mothers were born in Australia and New Zealand (*p*-value of interaction = 0.02, Appendix A). A similar pattern was seen for longitudinal eczema phenotypes up to 25 years (aMOR = 0.33, 95% CI = 0.15–0.71, Appendix A). There was also some evidence that maternal prenatal smoking and paternal education modified the association with eczema phenotypes up to 25 years. Higher vitamin D levels were associated with an increased risk of the mid-onset persistent eczema phenotype in participants with smoking mothers (aMOR = 1.43, 95% CI = 1.01–2.04, Appendix A) and fathers with lower education (aMOR = 1.43, 95% CI = 1.01–2.03, Appendix A).

### 3.5. Eczema/Sensitization Phenotypes

We did not find strong evidence for the association between higher levels of neonatal 25(OH)D3 and risk of atopic eczema up to age 25 years (Appendix A). However, there were trends that increased neonatal vitamin D levels were associated with a reduced risk of atopic eczema at most ages, including 2, 12, and 25 years (Appendix A). We did not find any effect modifiers for this association, including atopy at each age.

## 4. Discussion

This is the first study to explore the association between neonatal serum vitamin D levels and eczema phenotypes up to 25 years of age. We observed that higher neonatal vitamin D levels were associated with a lower risk of early-onset persistent eczema, but higher odds of early-onset-resolving eczema up to 25 years of age. There was stronger evidence of an association between vitamin D and reduced risk of early-onset persistent eczema in children with mothers who were born outside Australia or New Zealand. Higher vitamin D levels were associated with an increased risk of the mid-onset persistent eczema phenotype in participants with non-smoking mothers and fathers with lower education. We also observed evidence that higher levels of vitamin D were associated with a reduced risk of prevalent eczema at age 2 in children of mothers who had a history of smoking, but not in children of non-smoking mothers.

We observed very high rates of low 25(OH)D3 levels in this cohort, which is similar to other studies from this region that used dried blood spot cards [34]. However, the concentrations of 25(OH)D3 were higher than the results of another Melbourne newborn cohort, which also measured the neonatal dried blood spot vitamin D level (*n* = 100, born between September 2003 and March 2004) [35]. This shows the consistency of low vitamin D levels in Guthrie card samples from the Melbourne area. As Melbourne is distant from the equator, it receives relatively less sunlight than most other state capitals of Australia. Coupled with this, a skin cancer prevention campaign during the recruitment period of this cohort may explain these low levels. The level of serum 25(OH)D3 in our cohort was lower than in another Victorian cohort, which used cord blood serum samples (median 32.5 nmol/L vs. 50.7 nmol/L in the Barwon Infant Study (BIS) cohort) [36]. This cohort was recruited from a rural and coastal setting, where they may have had more sunshine exposure and therefore higher vitamin D levels. 

In our study, we did not find an association between vitamin D and prevalent eczema at different ages up to 25 years. However, our systematic review found consistent evidence that higher cord blood vitamin D levels were associated with a reduced risk of eczema [17]. Although the 25(OH)D3 concentration in neonatal cord serum and dried blood spots are highly correlated (r = 0.85) [35], we cannot directly compare our results with results from studies using cord blood vitamin D. Factors like blood spot volume, hole punch position, and paper type could significantly influence the analysis of DBS 25(OH)D3 [32].

One of our key findings was the association between neonatal vitamin D and reduced risk of early-onset persistent eczema, but increased odds of early-onset-resolving eczema, which has not been demonstrated previously. Eczema is typified by a relapsing/remitting course, and early-onset persistent eczema is the most burdensome subclass [28]. This suggests that higher vitamin D levels in neonatal life, a reflection of maternal vitamin D during pregnancy, may prevent children from developing early-onset persistent eczema. In contrast, we found that children who developed eczema and had higher vitamin D levels were more likely to have their eczema resolved (early-onset-resolving eczema), highlighting the value of longitudinal eczema phenotypes. The underlying etiology of early-onset-resolving eczema remains unclear, but it may be a group of early-life skin conditions that are not “true atopic eczema”. The initial eczema definitions for the first two years were mainly based on the doctor’s diagnosis or the use of topical steroids, while later eczema definitions also included parent/self-reported eczema episodes. The heterogeneity of the definitions might lead to a higher proportion of early-onset persistent eczema and subsequently less early-onset-resolving eczema. Further studies are required to confirm this finding, as it is not known why vitamin D levels would increase the risk of these transitory skin conditions. 

The protective association between neonatal vitamin D and early-onset persistent eczema suggests that vitamin D supplementation during pregnancy might reduce the risk of this outcome. Neonatal vitamin D level is a reflection of maternal vitamin D status, and supplementation during pregnancy increases neonatal vitamin D [18,19]. Current guidelines [24,25,26] do not recommend the routine use of vitamin D supplementation during pregnancy. To date, four RCTs have tested the impact of maternal supplementation on eczema outcomes in the child, ranging from 151 to 707 participants [20,21,22,23]. Collectively, these trials yielded a pooled estimate showing a weak protective effect on eczema (OR = 0.85, 95% CI = 0.67–1.08) [17]. None of these studies have examined longitudinal eczema phenotypes, and two trials [20,21] included supplementation of the control group with 400 IU of vitamin D, potentially masking any effects of this intervention. The one large study that did not contaminate the control group in this way showed a substantial reduction in risk of eczema in the child at 1 year of age (OR = 0.57, 95% CI = 0.33–0.98) [22]. As such, while there is insufficient evidence to support maternal supplementation with vitamin D during pregnancy at this time, given that vitamin D is a relatively cheap and safe intervention, further large and well-designed clinical trials are still needed.

We observed evidence that maternal birthplace modified the association between neonatal vitamin D and eczema phenotypes. Mother’s birthplace is a crude marker of potential differences in variance in vitamin D pathway genes, and also possibly a range of behavioral differences (both diet and sun exposure), that may impact these associations. The prevalence of eczema and vitamin D levels in Australia varies by ethnicity [37]. While one study in Melbourne has shown that vitamin D-deficient infants with Australian-born parents were more likely to be peanut/egg allergic, it did not find any effect modification by place of parents’ birth for the association between vitamin D and eczema [38]. Unfortunately, we only had a few participants whose mothers were born in Asia, so we could not directly test if our results varied by ethnicity. Further studies are needed, to test for heterogeneity of these associations by ethnicity and variations in vitamin D pathway genes. 

The association between neonatal vitamin D and the reduced risk of prevalent eczema at 2 years of age was stronger among children of ever-smoking mothers. Maternal prenatal smoking exposure has previously been found to be associated with an increased risk of atopic eczema [39]. It is unclear why higher levels of vitamin D were associated with an increased risk of mid-onset persistent eczema only in smoking mothers. Potential confounding by underlying socio-demographic or genetic factors may have generated this association. As smoking is a modifiable factor, this association should be investigated in future studies. 

This study has several important strengths but also some limitations. First, the prospective assessment of these associations in the study reduced the possibility of reverse causation. We planned our analysis to control for a range of confounding factors. Pollen and viral exposures in early life may influence the development of allergic diseases [40,41,42], but the evidence remains unclear. As we did not have direct measures of pollen and viral exposures, to block potential confounding through this pathway, we performed sensitivity analyses that adjusted for the season of birth as a proxy measure and also investigated the potential effect of the season of birth on the associations. This did not materially alter the associations observed with early-onset persistent eczema. 

One of the limitations of this study was the relatively small sample size and attrition over time, impacting the precision of some estimates. Nevertheless, the baseline characteristics and eczema status at each time point between the participants with or without vitamin D measures did not have any substantial differences, which suggests a low risk of selection bias. Furthermore, consistent trends were observed and reported accordingly. The measured absolute levels of 25(OH)D3 may be reduced due to sample degradation during prolonged storage. However, this would not impact the associations between vitamin D and eczema. This was a high-allergy-risk cohort, so our findings might not be generalizable to the general population. As we have performed multiple comparisons, some findings may be spurious. However, we only interpreted the findings based on the patterns of associations and not individual results. 

## 5. Conclusions

In summary, these findings suggest that higher levels of vitamin D in the neonatal period may reduce the risk of early-onset persistent eczema. Maternal smoking during pregnancy and birthplace may modify these associations. Observational studies, in more ethnically diverse populations, are required to confirm these findings. Given the limitations of the existing clinical trials, further well-designed, and adequately powered, clinical trials are needed to examine the impact of maternal vitamin D supplementation on the prevention of eczema, including early-onset persistent eczema. 

## Figures and Tables

**Figure 1 nutrients-16-01303-f001:**
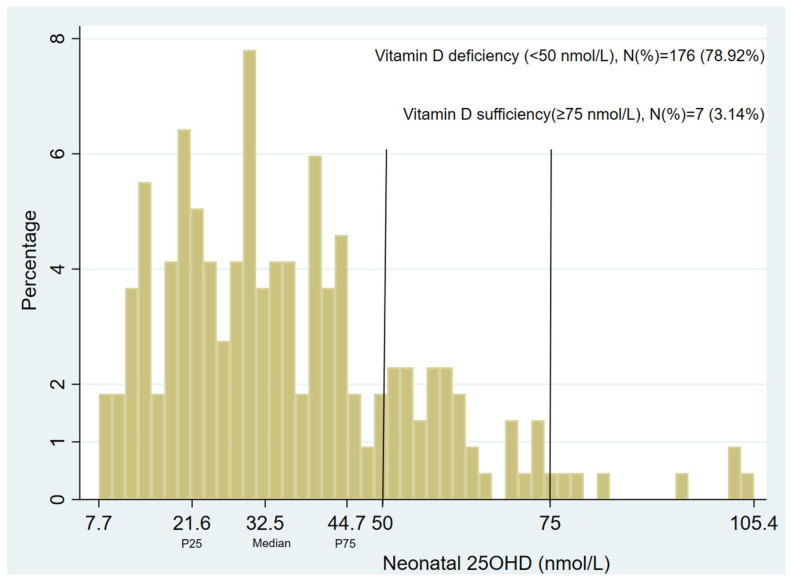
The neonatal vitamin D (25(OH)D3) level in 223 MACS participants. The median level was 32.5 nmol/L. P25 was 21.6 nmol/L, P75 was 44.7 nmol/L, and mean was 35.9 ± 18.6 nmol/L.

**Table 1 nutrients-16-01303-t001:** The associations between neonatal vitamin D (per 10 nmol/L increase) and prevalent eczema from birth to 25 years of age.

Estimates	Prevalent Eczema Age					
	1 Year	2 Years	6 Years	12 Years	18 Years	25 Years
n/N ^†^	70/220	63/214	26/174	25/155	47/196	32/122
Unadjusted associations						
OR (95% CI)	1.05 (0.90–1.22)	0.86 (0.72–1.02)	0.86 (0.67–1.10)	0.89 (0.69–1.14)	1.13 (0.99–1.28)	1.04 (0.88–1.22)
*p*-value	0.539	0.087	0.239	0.361	0.067	0.648
Adjusted associations (Model 1) ^‡^						
OR (95% CI)	1.06 (0.90–1.24)	0.86 (0.72–1.03)	0.87 (0.67–1.12)	0.91 (0.71–1.17)	1.03 (0.85–1.26)	0.94 (0.74–1.20)
*p*-value	0.489	0.091	0.275	0.484	0.741	0.623
Additionally adjusted for the season of birth (Model 2) ^§^						
OR (95% CI)	1.04 (0.87–1.24)	**0.80** **(0.65–0.98)**	0.81 (0.60–1.08)	0.86 (0.65–1.14)	1.01 (0.81–1.27)	0.87 (0.67–1.14)
*p*-value	0.658	**0.032**	0.155	0.301	0.909	0.318

^†^ N: the total number of participants with vitamin D and eczema outcomes; n: of these, the number who had eczema. ^‡^ Adjusted for dog ownership, maternal eczema history, mother’s age at delivery, maternal smoking during pregnancy, any siblings, mother’s birthplace, and maternal and paternal education. ^§^ Adjusting for all the variables in Model 1 and the season of birth. The pooled 1–25 years eczema result was not reported as the *p*-value for interaction with time at age 2 was <0.1 (*p* for interaction with time = 0.03).

**Table 2 nutrients-16-01303-t002:** The association between neonatal vitamin D and longitudinal eczema phenotype up to 12 years of age.

Estimates	Longitudinal Eczema Subclasses				
	Early-Onset Persistent	Early-Onset- Resolving	Mid-Onset Persistent	Mid-Onset- Resolving	Minimal/No Eczema
n/N ^†^	22/223	18/223	25/223	4/223	154/223
Unadjusted associations					
MOR (95% CI)	**0.75 (0.57–0.98)**	1.22 (0.98–1.52)	0.98 (0.81–1.18)	0.73 (0.40–1.31)	1.00 (Reference)
*p*-value	**0.039**	0.073	0.828	0.282	-
Adjusted associations (Model 1) ^‡^					
aMOR ^¶^ (95% CI)	**0.74 (0.56–0.98)**	**1.30 (1.05–1.62)**	0.97 (0.79–1.19)	0.66 (0.35–1.24)	1.00 (Reference)
*p*-value	**0.036**	**0.016**	0.787	0.196	-
Additionally adjusted for the season of birth (Model 2) ^§^					
aMOR (95% CI)	0.75 (0.55–1.02)	1.21 (0.93–1.56)	0.89 (0.70–1.13)	0.68 (0.38–1.23)	1.00 (Reference)
*p*-value	0.068	0.150	0.351	0.206	-

^†^ N: the total number of participants with vitamin D and eczema outcomes; n: of these, the number who had eczema. ^‡^ Adjusting for dog ownership, maternal eczema history, mother’s age at delivery, maternal smoking during pregnancy, any siblings, mother’s birthplace, and maternal and paternal education. ^§^ Adjusting for all the variables in Model 1 and the season of birth. ^¶^ aMOR: adjusted multinomial odds ratio.

**Table 3 nutrients-16-01303-t003:** The association between neonatal vitamin D and longitudinal eczema phenotypes up to 25 years of age.

Estimates	Longitudinal Eczema Subclass			
	Early-Onset Persistent	Early-Onset-Resolving	Mid-Onset Persistent	Minimal/No Eczema
n/N ^†^	21/223	20/223	31/223	151/223
Unadjusted associations				
MOR (95% CI)	0.76 (0.58–1.01)	1.13 (0.89–1.43)	1.03 (0.85–1.24)	1.00 (Reference)
*p*-value	0.055	0.305	0.750	-
Adjusted associations (Model 1) ^‡^				
aMOR ^¶^ (95% CI)	0.76 (0.58–1.01)	1.18 (0.93–1.50)	1.03 (0.84–1.25)	1.00 (Reference)
*p*-value	0.057	0.170	0.780	-
Additionally adjusted for the season of birth (Model 2) ^§^				
aMOR (95% CI)	0.76 (0.55–1.04)	1.05 (0.86–1.42)	0.96 (0.75–1.22)	1.00 (Reference)
*p*-value	0.088	0.444	0.730	-

^†^ N: the total number of participants with vitamin D and eczema outcomes; n: of these, the number who had eczema. ^‡^ Adjusting for dog ownership, maternal eczema history, mother’s age at delivery, maternal smoking during pregnancy, any siblings, mother’s birthplace, and maternal and paternal education. ^§^ Adjusting for all the variables in Model 1 and the season of birth. ^¶^ aMOR: adjusted multinomial odds ratio.

## Data Availability

Data from this study are not available, as participants of this study did not give written consent for their data to be shared publicly.

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
