# Peer review of "Neonatal Vitamin D and Associations with Longitudinal Changes of Eczema up to 25 Years of Age"

_nutrients, 2024, doi:10.3390/nu16091303_

Round 1
Reviewer 1 Report
Comments and Suggestions for Authors
I just finished reading a very interesting study regarding vitamin D levels in newborns and its association to eczema in childhood and early adulthood.
The introduction can be improved, adding more information regarding the pathophysiology of eczema and the role of vitamin D in Th2 inflammation.
The Methods are excellently presented. Regarding the Ethics, I believe you should add the protocol numbers and dates of the approvals.
The discussion is in accordance with the presented results.
You mention that prenatal smoking exposure has previously been found to be associated with an increased risk of atopic eczema. Is there any pathway described or suggested?
Dear authors, you conducted a very well-designed cohort study, while you presented its results appropriately.
Comments on the Quality of English Language
No language editing is needed.
Reviewer 2 Report
Comments and Suggestions for Authors
Nicely presented study, no major concerns.
Were cord vitamin D levels available for comparison?
Comments on the Quality of English Language
Well written.
Reviewer 3 Report
Comments and Suggestions for Authors
The study raises a very interesting issue regarding eczema risk factors in long-term follow-up. I have no objections to the way in which the analyzed factors are correlated, but I wonder whether the determination of vitamin D using the dried blood spots method is already considered reliable in science. The method (formula) for converting vitamin D concentration should also be explained more clearly (line 91).
Other comments:
- I don't understand Figure No. 1. What does "density" mean on the vertical axis?
- the description of some results is not understandable. Once the authors write that there was no relationship between the concentration of vitamin D and eczema, while elsewhere they write that a higher concentration of vitamin D was associated with a lower risk of early-onset-persistent eczema
- instead of "we did not find strong evidence" it should simply be "we did not find evidence", i.e. without the word “strong” (lines 230-231) ??
- in lines 225-228 the authors write "Higher vitamin D levels were associated with an increased risk of the mid-onset-persistent eczema phenotype in participants with non-smoking mothers...", while in the discussion it is stated "It is unclear why higher levels of vitamin D were associated with increased risk of mid-onset-persistent eczema only in the smoking mothers” (lines 313-314) – in the last sentence should probably be “non-smoking” ?
- table numbers are incorrect throughout the article. Instead of Table S2 there should be Table S1, instead of Table S3 there should be Table S2, etc. Table No. 9, which the authors write about, does not exist at all
- in line 218 it is stated that there were 23 mothers born outside Australia and New Zealand, while in tables S4 and S5 it is written that there were 28 mothers
- in Table S1, the error in the newborns' body weight should be corrected (34678 ± 509 g)
- table S2 does not indicate which values were statistically significant (**)
